# *In vivo* evaluation of phage therapy against *Klebsiella pneumoniae* using the *Galleria mellonella* model and molecular characterization of a novel *Drulisvirus* phage species

Gustavo Quispe-Villegas,[1,2] Gabriela I. Alcántara-Lozano,[1,2] Diego Cuicapuza,[1,2,3] Raúl Laureano,[1,4] Brenda Ayzanoa,[3] Pablo Tsukayama,[3,5,6] Jesús Tamariz[1,2]

**ABSTRACT** Multidrug-resistant (MDR) *Klebsiella pneumoniae* is challenging to treat with conventional antibiotic regimens, posing a threat to healthcare systems. Phage therapy presents a promising alternative treatment strategy; however, characterization of its efficacy and safety is required. Here, we describe the microbiological and molecular characterization of a novel bacteriophage with activity against MDR *K. pneumoniae* using a greater wax moth (*Galleria mellonella*) model system. A bacteriophage was isolated from hospital wastewater. Viral kinetics and phage stability were evaluated under varied pH and temperature conditions. The therapeutic efficacy of the phage was evaluated using MDR *Klebsiella*-infected *G. mellonella* larvae as an *in vivo* model. Phage titers and larva survival were compared in phage-treated and control groups. Genomic sequencing (Nanopore and Illumina) was used to classify the bacteriophage and identify any resistance genes or virulence factors present in its genome. Functional characterization demonstrated effective lytic activity, favorable burst size (161 PFU/cell), and an optimal MOI of 0.1. The phage demonstrated stability across a wide range of temperatures (8°C–40°C) and pH levels (4–8). Experiments using the *G. mellonella* model showed improved larval survival with phage treatment. The novel bacteriophage was identified as a new species within the genus *Drulisvirus* with no lysogeny-associated, antimicrobial resistance, or virulence genes detected. The new *Drulisvirus* phage identified is a promising candidate for treatment of infections caused by MDR *K. pneumoniae*.

**IMPORTANCE** The study describes a bacteriophage with potential for use in phage therapy against *Klebsiella pneumoniae*, one of the most clinically significant bacterial pathogens today. Microbiological and genomic characterization of the phage revealed advantageous properties for therapeutic applications, while also identifying a novel species within the *Drulisvirus* genus. These findings significantly contribute to our understanding of bacteriophage diversity and their utility in combating antibiotic-resistant infections. Moreover, the authors developed an *in vivo* preclinical model of MDR infection using *Galleria mellonella* larvae and successfully applied it to study the bacteriophage's therapeutic efficacy. This model offers a robust and efficient platform for preclinical testing.

**KEYWORDS** bacteriophages, phage therapy, *Klebsiella pneumoniae*, *Galleria mellonella*

*K*lebsiella pneumoniae is a Gram-negative bacterium in the *Enterobacterales* order and is recognized as one of the most important opportunistic pathogens, responsible for nosocomial infections including soft tissue infections, urinary tract infections, intra-abdominal infections, pneumonia, bacteremia, septicemia, and septic shock (1–3). Due to nosocomial infections, the use of antibiotics over several decades has resulted in the

**Peer Reviewers** Ayaz Ahmed, International Center for Chemical and Biological Sciences, University of Karachi, Karachi, Pakistan; Nana Ama Amissah, Noguchi Memorial Institute for Medical Research, University of Ghana, Legon, Accra, Ghana

Address correspondence to Jesús Tamariz, jesus.tamariz@upch.pe.

The authors declare no conflict of interest.

emergence of multidrug-resistant (MDR) strains of *K. pneumoniae* (4, 5). Infections caused by MDR *K. pneumoniae* contribute to increased mortality, worse clinical outcomes, prolonged hospital stays, inappropriate overuse of antibiotics, and are a persistent threat to healthcare systems (2, 5, 6). It is estimated that in 2019, 4.95 million deaths associated with resistant bacteria occurred worldwide, with *K. pneumoniae* included in the list of pathogens (7). Additionally, according to the World Bank, antimicrobial resistance is projected to cost up to $6.1 billion annually worldwide by 2050 (8).

Because of the high morbidity and mortality associated with MDR *K. pneumoniae*, phage therapy has regained attention as an alternative treatment for infections caused by antibiotic-resistant bacteria (9–12). Several studies have reported the application of phage therapy in severe infections, such as curative treatment for patients with recurrent urinary tract infections caused by MDR *K. pneumoniae* (13, 14). However, not all phages are suitable for therapeutic purposes; thus, identification and characterization are fundamental initial stages for any phage-related research (12, 15). Those with lytic replication cycles are preferred because of a lower risk of horizontal gene transfer (HGT) to pathogens, which may include virulence factors and antibiotic resistance genes (15, 16).

Furthermore, the evaluation of *in vivo* efficacy has been improved with the introduction of the *Galleria mellonella* model, an insect of the order *Lepidoptera*, which exhibits, during its larval stage, a range of characteristics that make it a suitable model: economical, easy to maintain, and requiring no specialized laboratory equipment (17–19). Additionally, it represents an alternative to murine models for the study of microbial infections without the disadvantages of high maintenance costs and long reproduction time. The ethical scope and reduced biological complexity of *Galleria spp.* make it an alternative model for *in vivo* assessment of the virulence of Gram-negative bacteria (19, 20). Based on the above, the present study was conducted with the aim to isolate and characterize local phages against an MDR *K. pneumoniae* isolate and assess *in vivo* efficacy using the *G. mellonella* model.

## MATERIALS AND METHODS

### Bacterial isolates

The isolate of *K. pneumoniae* utilized in this study was an extended-spectrum beta-lactamase (ESBL)-producing strain, obtained from the biobank of the Laboratorio de Resistencia a Antimicrobianos y Fagoterapia. Seven more carbapenemase-producing *K. pneumoniae* isolates were also obtained for the host range assay. All isolates were cryopreserved in Tryptic Soy Broth (TSB) with 20% glycerol at −20°C. Bacteria for experimentation were cultured on Luria Bertani (LB) agar (LB with 1.5% weight/volume agar) at 37° for 24 h. The aforementioned ESBL strain was used for local phage isolation, characterization, and *in vivo* experiments with the *G. mellonella* model; the carbapenemase-producing *K. pneumoniae* strains were only used in the host range assay.

### Phage isolation

Phages were isolated from wastewater collected from the drainage of a tertiary hospital in Lima, Peru. A sample of 10 mL was obtained and subsequently subjected to filtration using 0.22 µm pore-size syringe filters to remove debris and bacteria. Filtration was repeated three times. For enrichment, 200 µL of bacterial solution in 0.9% normal saline with a turbidity of 0.5 McFarland units was inoculated into 1 mL of filtered wastewater in 90 mL of double concentrated Luria Bertani (LB) broth, which was then incubated overnight at 37°C in a shaker at 110 rpm for 24 h. Subsequently, three filtrations were performed using 0.22 µm pore-size syringe filters to remove cells. This filtrate was serially diluted to $10^{-4}$ and $10^{-6}$ in 0.9% normal saline.

An overlay agar plaque assay was conducted: a mix of 100 µL bacterial solution in 0.9% normal saline with a turbidity of 0.5 McFarland units was inoculated with 100 µL of

each diluted viral filtrate, then incubated at room temperature for 5 min and mixed with 5 mL of molten top LB-semisolid agar (LB with 0.4% weight/volume agar). The mix was poured onto LB agar plates. Overnight incubation was performed on inverted plates at 37°C, which were examined the next day to identify lytic plaques.

Single plaques with clear lytic halos were picked and inoculated in 1.5 mL of LB broth. The solution was then incubated at 37°C for 1 h to allow the bacteriophage particles to diffuse out of the agar, as described in a previous protocol (21), followed by centrifugation for 2 min at 1500 rpm. The supernatant was collected, and the filtrate was subjected to three subsequent rounds of plaque assay for the isolation and expansion of clonal phages. Purified phages were then stored at 8°C.

## Phage–host range assay

In addition to the bacterial strain from which the phages were isolated, the host range was determined against six different clinical carbapenemase-producing *K. pneumoniae* isolates (VIM-1, IMP-8, NDM-1, KPC-2, OXA-48-like, and OXA-181) by standard spot assays. Bacterial cultures were suspended in 0.9% normal saline and titrated to a 0.5 McFarland scale. Aliquots of 5 µL of diluted phage lysate ($10^8$ PFU/mL) were spotted on bacterial lawns over Luria Bertani (LB) agar. After incubating at 37°C for 24 h, the spots were assessed by the formation of lytic plaques without turbidity for positive infection. The experiment was performed as independent duplicates.

## Phage stability at different temperature and pH ranges

For pH testing, SM buffer (50 mM Tris-HCl, 8 mM magnesium sulfate, 100 mM sodium chloride, 0.01% gelatin, pH 7.5) was prepared and adjusted to pH values of 4, 5, 6, 7, and 8 in sterile test tubes. Then, a viral solution of $10^8$ PFU/mL was added in a 1:10 ratio with SM buffer. The mixture was incubated at 37°C in a shaker at 210 rpm for 1 h. For temperature stability, a viral solution in 0.9% normal saline was added in a 1:10 ratio, followed by incubation at 8°C, 35°C, 36°C, 37°C, 38°C, 39°C, 40°C, and 50°C for 1 h. For both experiments, lytic activity was evaluated with the aforementioned agar plaque assay.

## Optimal multiplicity of infection (MOI) and one-step growth curve

The multiplicity of infection (MOI) is the ratio between the plaque forming units (PFU/mL) and the colony forming units (CFU/mL). A bacterial solution in 0.9% normal saline was obtained from a log-phase culture of the bacterial isolate (OD600 = 0.6), and was mixed with viral filtrate to obtain samples with an MOI of 0.1, 1, or 10. The mixture was cultivated at 37°C for 4 h, and phages were filtered using 0.22 µm pore-size syringe filters. Progeny phages were quantified using an agar plaque assay performed in duplicate. The optimal MOI was considered to be the one that presented the highest PFU/mL.

For the one-step growth curve, a mixture of bacteria and phage in 0.9% normal saline at the optimal MOI was generated as described above. Subsequently, the mixture was incubated for 15 min at 37°C in a shaker at 100 rpm, followed by centrifugation at 3,000 rpm in 2 cycles of 10 min each. The supernatant was separated, and the pellet was resuspended in 1 mL of LB broth. Then, 100 µL of the obtained solution was added to 50 mL of LB broth tempered at 37°C. This was considered as time zero, from which duplicate samples of 1 mL of broth were taken every 10 min for a total duration of 110 min.

These samples were utilized for agar plaque assays to determine viral concentration via serial dilutions spanning $10^{-2}$ to $10^{-8}$.

## *Galleria mellonella in vivo* infection model

Larvae of *G. mellonella* without dark spots and with a longitudinal length greater than 25 mm were selected. Decontamination was performed using a sterile cotton swab

soaked in 70% ethanol. All bacterial inoculations were carried out using sterile 300 µL insulin syringes into the last left posterior proleg. Viral inoculations were administered into the last right posterior proleg. The larvae were then incubated in Petri dishes at 35°C in darkness and were assessed every 24 h for a total of 96 h. Larvae were considered dead if there was a loss of spontaneous movement without response to tactile stimulation using sterile 100 µL pipette tips.

A bacterial solution in 0.9% saline (OD600 = 0.6) was prepared and diluted to $10^7$ CFU/ml with SM buffer. Concomitantly, the viral lysate was diluted with buffer SM until reaching the concentration needed to match the empirically determined optimal MOI described above.

A cohort of 20 randomly chosen larvae was distributed among four groups of five larvae each: a control group was inoculated with 200 µL of SM buffer; a second group received an inoculum of 200 µL of bacterial suspension; a third group received 200 µL of viral concentrate; and a fourth group was inoculated with a mixture of bacterial and viral suspensions in a 1:1 ratio (200 µL each). For the last group, the bacterial suspension was initially inoculated, and after 60 min, the bacteriophage suspension was added. Experiments were performed independently in triplicate.

## Whole genome sequencing (WGS) and bioinformatic analysis

Phage DNA was extracted using a ZymoBIOMICS DNA Miniprep Kit (Zymo Research), according to manufacturer's instructions. Phage DNA libraries were prepared with an Oxford Nanopore rapid PCR barcoding kit using 10 ng of DNA and sequenced on a R10.4.1 flow cell with a MinION Mk1B sequencer (Oxford Nanopore Technologies). Additionally, Illumina libraries were prepared using the Illumina DNA Prep kit and sequenced on a NovaSeq X Plus (Illumina), producing paired-ended reads of 151 bp.

DNA adapters were removed from raw nanopore reads using Porechop v0.2.3 (https://github.com/rrwick/Porechop). Subsequently, processed reads were aligned to a *K. pneumoniae* reference genome (strain MGH 78578, accession number CP000647.1) using Minimap2 v2.22 (https://github.com/lh3/minimap2); potential contaminants were eliminated.

Unaligned reads were pooled into new FASTQ files using Samtools v1.13 (https://github.com/samtools/samtools), and subsequently, *de novo* assemblies were generated using Flye v2.8.1 (https://github.com/mikolmogorov/Flye) with the "--meta" option. Preliminary assemblies were polished with the Illumina reads using Polypolish v0.6.0 (https://github.com/rrwick/Polypolish) and reoriented with Dnaapler v0.8.0 (https://github.com/gbouras13/dnaapler).

Antimicrobial resistance genes were identified using AMRFinderPlus v3.12.3 (https://github.com/ncbi/amr; Database version: 2024–07-22.1), and virulence factors were detected using Abricate v1.0.1 with the Virulence Factor Database (VFDB) (https://github.com/tseemann/abricate; Database version: 2024-Nov-26).

Annotations were conducted using the curated phage database phold v0.2.0 (https://github.com/gbouras13/phold). Taxonomic identification was performed with tax_myPHAGE (https://github.com/amillard/tax_myPHAGE?tab=readme-ov-file) and PhaBOX2 (https://github.com/KennthShang/PhaBOX?tab=readme-ov-file).

## Statistics

To compare whether phage titers differed significantly between the aforementioned temperature and pH ranges, a Kruskal–Wallis test was performed. Subsequently, if significant differences were found, post-hoc Dunn's test was conducted to determine which specific conditions differed.

With the larval survival results over the 96 h period, a Kaplan–Meier curve was constructed, and the log-rank test was performed to compare survival curves. All statistical tests were carried out at a significance level of $P < 0.05$. All analyses and

graphics were performed using R-Studio software v4.2.2 and the statistical packages ggplot2 v3.3.4, survival v3.5–7, and dplyr v1.1.4.

## RESULTS

### Isolation of bacteriophage *GA23* and host range determination

Hospital wastewater incubated with the ESBL-producing *K. pneumoniae* isolate has successfully yielded multiple bacteriophages capable of forming lytic halos, as demonstrated by the agar plaque assay method. Three phages with the biggest plaque diameters (9 mm) were selected. However, subsequent passages for enrichment and isolation of a clonal virus resulted only in one representative isolate, GA23, with sufficient titer (>$10^8$ PFU/mL) for isolation, characterization, and *in vivo* assays. Around the clean central plaque, the formation of a secondary semi-transparent halo was observed (Fig. 1).

When tested against six different carbapenemase-producing *K. pneumoniae* strains (VIM-1, IMP-8, NDM-1, KPC-2, OXA-48-like, and OXA-181), GA23 exhibited no bacteriolytic activity and failed to infect either strain, aside from the ESBL-producing *K. pneumoniae*.

### pH and temperature stability of GA23

Phage GA23 formed clear lytic halos under varied pH conditions ranging from pH 4 to pH 8 (Fig. 2). No difference in titer greater than the specificity of the assay (approximately 10-fold) was observed. No statistical differences between pH conditions were found ($P = 0.7546$). For thermal stability, the maximum titer was obtained at 8°C, and titers were relatively consistent between 35°C and 40°C. No plaques formed at a temperature of 50°C (Fig. 2). These differences were statistically significant across all temperature values ($P = 0.0009$), as well as when comparing between 8°C and 35°C ($P = 0.0016$) and between 40°C and 50°C ($P = 0.0195$).

### Optimal MOI and one-step growth curve of GA23

Maximum titers of phage GA23 were obtained when starting with a MOI of 0.1. One-step growth curve and the *in vivo* assay with *G. mellonella* experiments were conducted at this concentration. The one-step growth curve indicated that the exponential phase lasted approximately 30 min, with a burst size of 161 viral particles/cell (Fig. 3). No latent period was noted.

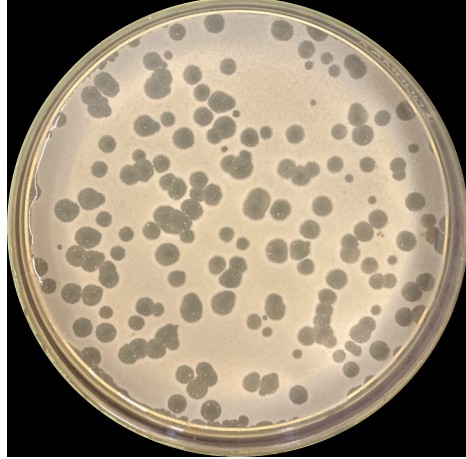

**FIG 1** Morphology of phage GA23 on LB agar. A translucent halo surrounds each lytic plaque after overnight culture.

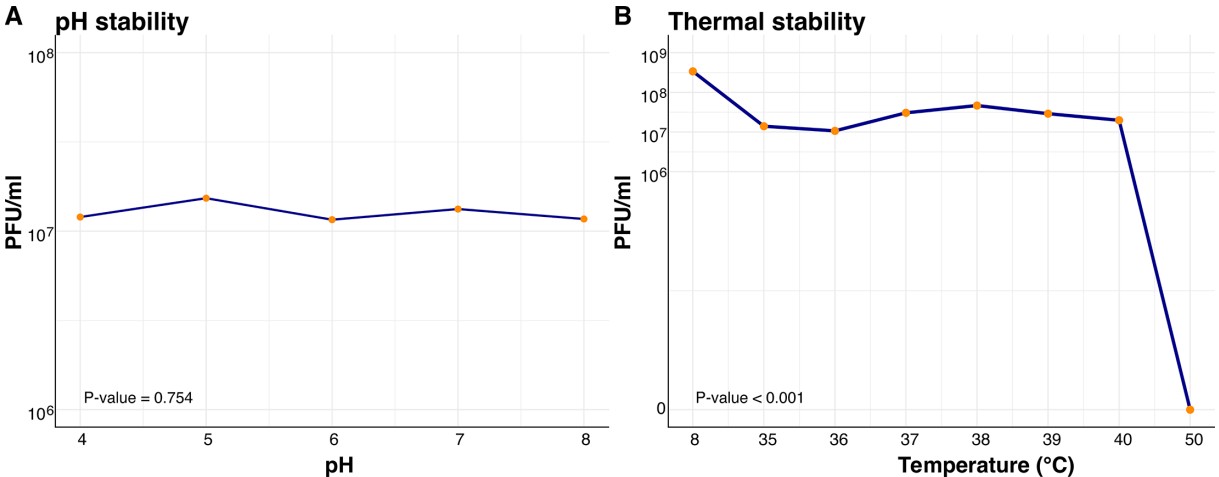

**FIG 2** (A) pH stability of phage GA23 after incubation at different pH for 1 h. (B) Thermal stability of phage GA23. Lytic activity was lost when GA23 was incubated at 50℃ for 1 h.

### *Galleria mellonella in vivo* infection model

In the *G. mellonella* model, the viability of the control group inoculated with bacterial solution was 53% on Day 1 and fell to 26.7% by Day 4. No mortality was observed among the groups treated with SM buffer or viral lysate. When phage GA23 was inoculated at an MOI of 0.1, 60 min after injection of the bacterial inoculum, viability of the larvae was 100% preserved at 96 h. Statistical analysis using the log-rank test revealed a significant difference in survival curves between the untreated control and phage-treated larvae ($P = 0.007$) (Fig. 4).

### Genome characterization of phage GA23

Phage GA23 has a 44,172 bp linear double-stranded DNA genome with a GC content of 53.7%. Fifty-eight open reading frames (ORFs) were identified in the complete genome, of which 31 (53.5%) were assigned known functions. Proteins related to phage structure, DNA packing, gene regulation, and cell lysis were identified. No transfer RNAs (tRNAs), transfer-messenger RNAs (tmRNAs), antimicrobial resistance genes, lysogenic factors, or virulence factors were detected. The complete phage genome circular map is shown (Fig. 5).

The DNA replication module includes 13 ORFs: DNA polymerase, DnaB-like replicative helicase, endonuclease VII, and DNA primase—all essential for DNA replication during the lytic cycle. The phage structure and packaging proteins module includes seven ORFs: head scaffolding protein, major head protein, an internal virion protein with an endolysin domain, tail fiber protein, and tail protein. Sequence analysis also identified an Rz-like spanin protein and phage endolysins.

Genome-based taxonomy revealed that GA23 belonged to the class *Caudoviricetes*, family *Autographiviridae*, subfamily *Slopekvirinae*, and genus *Drulisvirus*. The GA23 phage genome shares a nucleotide similarity greater than 70% with the genus *Drulisvirus* and less than 95% with species from its most closely-related cluster. Therefore, according to the International Committee on Taxonomy of Viruses (ICTV), it constitutes a new species, which we have named *Drulisvirus cayetanensis*.

### DISCUSSION

In this study, we recovered a novel lytic bacteriophage (named GA23) from a hospital sewage sample, using an MDR *K. pneumoniae* isolate as the target host. A narrow host range was observed, with no lytic activity on the six carbapenemase-producing *K. pneumoniae* isolates we tested. This could be attributed to the diverse spectrum of

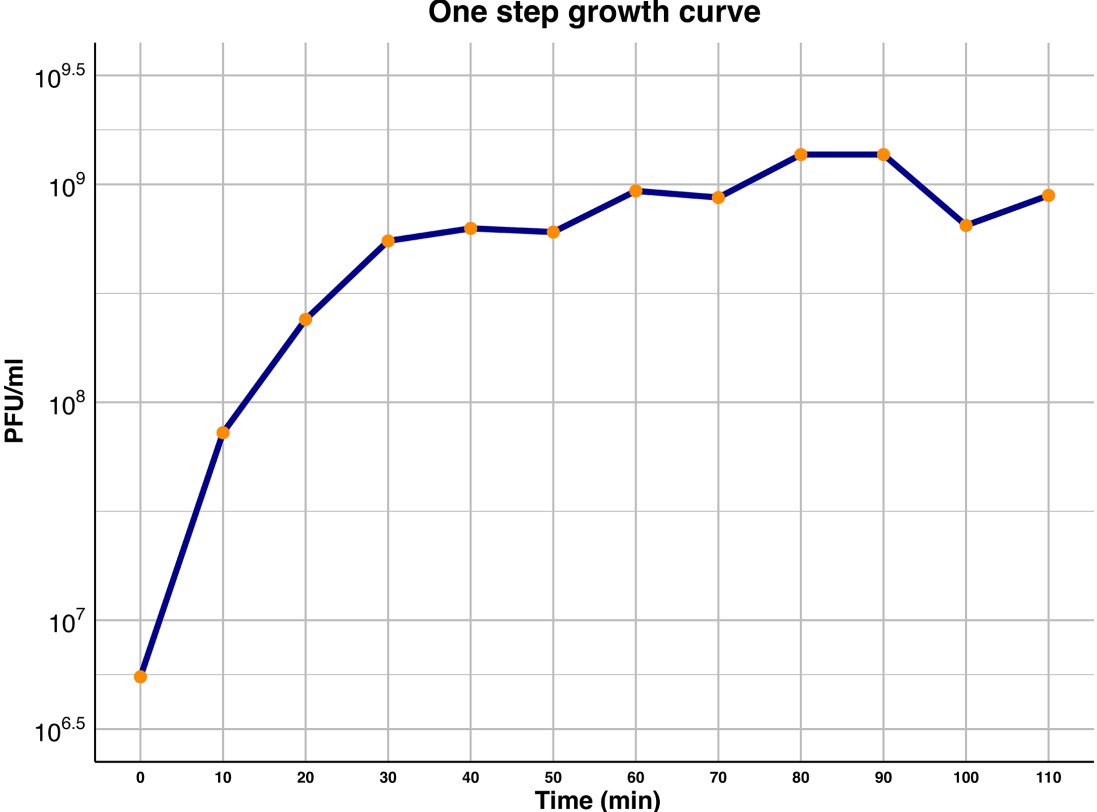

**FIG 3** One-step growth curve of phage GA23. Exponential phase lasted 30 min, with a burst size of 161 viral particles/cell.

capsular and membrane antigens in the *Klebsiella* genus, as well as the specificity of phage receptor-binding proteins (RBPs)(22). For instance, *K. pneumoniae* capsular antigens comprise a total of 79 K serotypes and 168 genotypes (23, 24). Even a single broad-spectrum bacteriophage is insufficient to target a large proportion of all described capsular types, especially when considering the additional variability introduced by non-capsule antigens (25, 26). Thus, it is difficult to isolate a phage that covers multiple clinical *K. pneumoniae* strains, as in the case of GA23. Although our isolate is specific to an ESBL host, this finding does not diminish its therapeutic potential: ESBL-producing *Enterobacteriaceae* contribute to higher rates of treatment failure, increased healthcare costs, and are recognized by the World Health Organization (WHO) as a critical priority group (27).

We observe a consistent phenomenon of secondary, semi-opaque halos around the central GA23 lytic plaque. This finding suggests the expression of phage depolymerases, enzymes with catalytic activity towards the capsule and bacterial peptidoglycan cell wall (28). Because depolymerases are soluble, translucent halos are formed from viable denuded bacteria at the periphery of the plaques (22, 24, 29, 30). Our WGS showed that GA23 possessed peptidoglycan lytic transglycosylases in both tail fiber and tail proteins, thus acting as depolymerases and RBPs. It is noteworthy that, although degradation of peptidoglycan results in destruction of bacterial cells, phage-encoded lytic transglycosylases can only cause local damage without lysis (31). For this reason, GA23 exhibits opaque clear halos and evidence of depolymerase activity.

Isolated phages intended for clinical use should remain stable in harsh microenvironment conditions; otherwise, direct therapeutic applications may be limited (32, 33). GA23 maintains lytic activity at temperatures ranging from 8°C to 40°C. This is relevant for phage storage through refrigeration, as well as for treatment of warm-blooded mammals like humans, under both physiological and febrile conditions (34). Incubation at 50°C

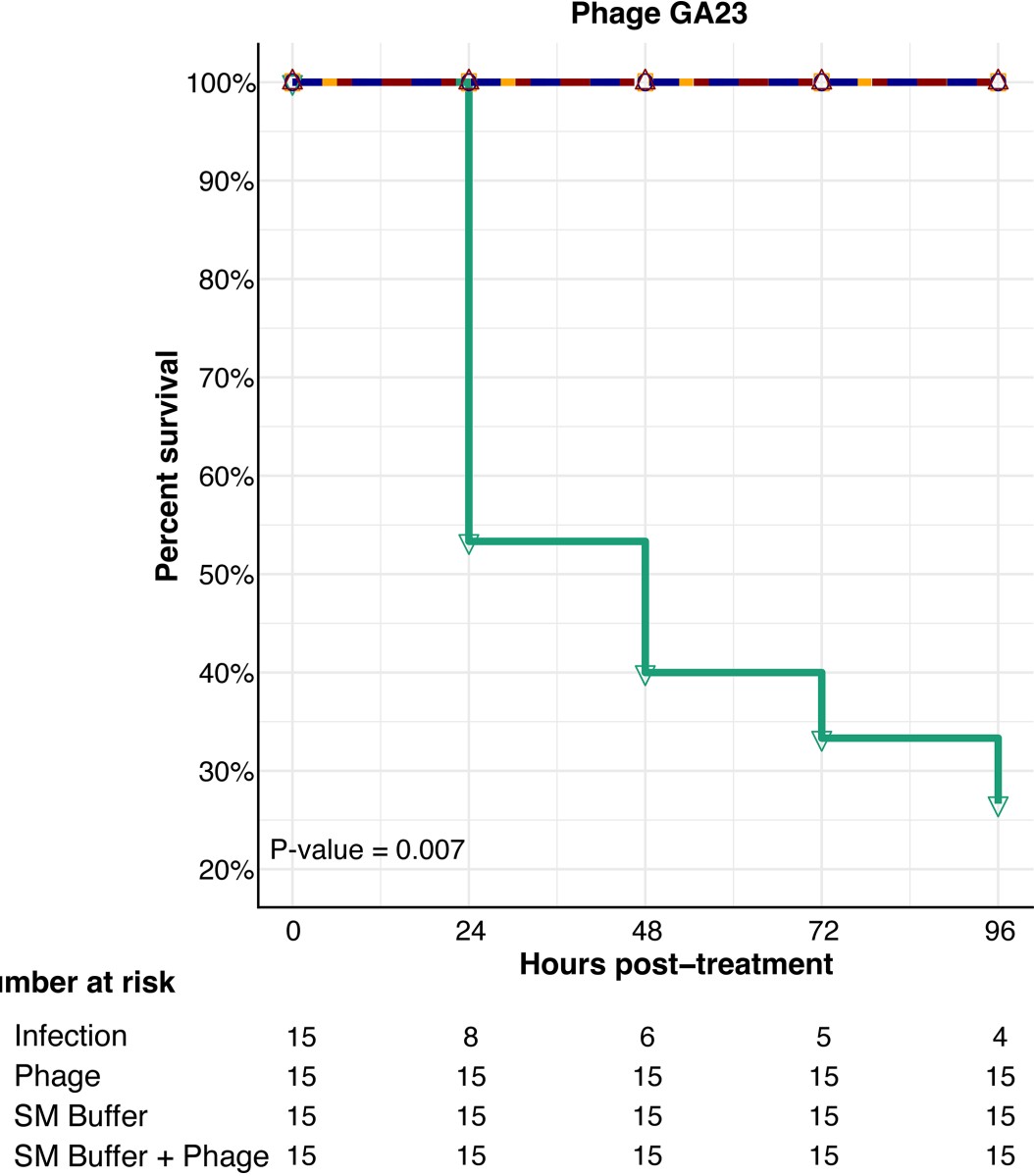

**FIG 4** Percentage of survival of *G. mellonella* larvae infected with ESBL-producing *K. pneumoniae*. Each group comprised 15 larvae. Notably, groups inoculated with phage, SM-buffer, or both maintained 100% survival. Only 26.7% of non-treated larvae survived after 96 h.

completely inactivated the phage; this temperature is lower than some other bacterio-lytic phages, some of which can tolerate temperatures as high as 60°C–70°C (13, 35–37). Nevertheless, this feature does not restrict clinical applicability, as these values are nearly incompatible with human life.

In contrast, a large range of pH values can be found in microenvironments within the same organism under physiological conditions. For example, human microbiota is responsible for the skin's alkalinity, while the lungs, kidneys, and urinary tracts have dynamic acid–base values (32, 38). GA23 achieved stability at a pH range of 4–8, without a significant change in viral titers between all conditions assessed. This is particularly relevant for *K. pneumoniae* phage therapy, as pneumonia, urinary tract infections, and septic shock can all contribute to blood and parenchymal pH disturbances (39). Other reported isolates retained lytic activity at pH ranges of 4–12, which is similar to the pH stability of GA23 and supports its suitability for use in phage therapy (13, 37, 40).

# *Drulisvirus cayetanensis* (GA23)

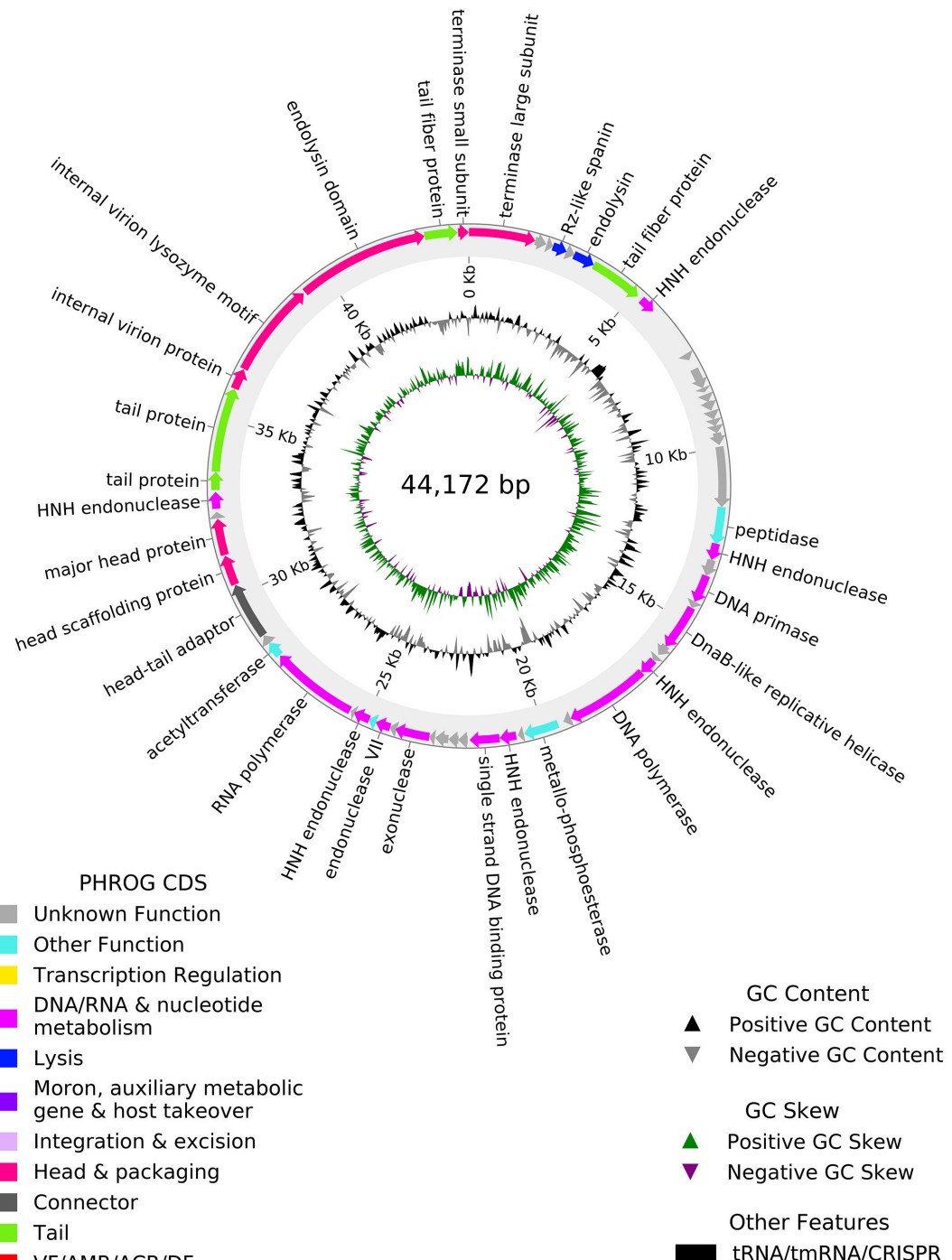

## PHROG CDS

- **Unknown Function** (gray)
- **Other Function** (cyan)
- **Transcription Regulation** (yellow)
- **DNA/RNA & nucleotide metabolism** (magenta)
- **Lysis** (blue)
- **Moron, auxiliary metabolic gene & host takeover** (purple)
- **Integration & excision** (light purple)
- **Head & packaging** (pink)
- **Connector** (dark gray)
- **Tail** (green)
- **VF/AMR/ACR/DF** (red)

### GC Content
- ▲ Positive GC Content
- ▼ Negative GC Content

### GC Skew
- ▲ Positive GC Skew
- ▼ Negative GC Skew

### Other Features
- ■ tRNA/tmRNA/CRISPR

**FIG 5** Circular genome annotation of phage *Drulisvirus cayetanensis*. The inner rings show genome location. The most external rings show identified open reading frames.

The importance of MOI lies in optimizing the generation of new virions during each lytic cycle. This value differs between phage strains and is influenced by factors, such as viral and bacterial sizes, receptor availability, and bacterial cell division, which introduce new hosts as the viral load increases (35). Theoretically, higher values lead to a greater reduction in bacterial burden; however, MOI values ranging from 1,000 to 0.001 have been reported (32, 33, 41). *In vivo* assays conducted against *G. mellonella* should be

performed at an optimal MOI, as this can directly impact survival rates. For example, at an MOI of 100, phage vB_KpnP_k3_ULINTkp1 rescues 53% of the larvae at 72 h, whereas another strain (phage vB_KpnP_k3_ULINTkp2) only achieves a survival rate of 27% (36). Our isolate demonstrates an optimal MOI of 0.1, which maximized the production of virions for subsequent *in vivo* assays. Moreover, this value potentially enables the production of a large number of bacteriophages at a lower cost, as less lysate is required (42).

The one-step growth curve is essential for assessing viral kinetics, as many of its parameters influence plaque properties. For example, plaque size is predicted to be negatively correlated with the latent period; as a longer intracellular phase reduces the time available for progeny virions to diffuse to new host cells (43). GA23 demonstrates a very short latent period, with the first virions released as early as 5 min post-infection. This feature was reported with previous *Drulisvirus* isolates, with latent periods lasting between 5 and 10 min (36, 44). Additionally, GA23 can produce up to 161 virions per cell, a burst size consistent with values reported for other *Drulisvirus* phages, ranging from 80 to 303 virions per cell (44–46). While larger burst sizes are positively correlated with plaque size, simulations showed that there is a diminishing impact for extreme values; therefore, kinetics assessment must be performed for all isolates intended for clinical use (43).

Genomic characterization of GA23 revealed a linear dsDNA genome. This is consistent with other species of *Drulisvirus*, with genome lengths ranging from 44,000 to 45,000 bp and GC% contents of 53.6%–53.8%. Moreover, genome annotation predicted four gene modules: DNA replication, phage structure, packaging proteins, and cell lysis. The last module contains a molecular triad consisting of holin, endolysin, and spanin, a multigene lysis system found in the *Caudoviricetes* class (47, 48). Holins accumulate in the inner cell membrane and form holes after reaching a critical concentration, exposing peptidoglycan to endolysins for degradation. Subsequently, spanins target the outer membrane of Gram-negative bacteria, which serves as the final barrier for newly formed virions during cell lysis. Hence, this triad is necessary to achieve a complete lytic infection cycle (49, 50).

Temperate or lysogenic phages may stably integrate into the host genome, preventing cell lysis. They may also enter the lytic cycle and promote horizontal gene transfer of antimicrobial resistance genes or virulence factors. These traits are utterly incompatible for phage therapies (29, 33, 51). Furthermore, indirect evidence of transduction, such as the presence of tRNA and tmRNA genes, may indicate the presence of bacterial DNA and phage lysogeny genes (24, 52). Fortunately, none of the aforementioned lysogeny genes or markers were detected in the GA23 genome.

According to the classification system of the International Committee on Taxonomy of Viruses, the criterion for species demarcation of bacterial and archaeal viruses is 95% of genome similarity (53). GA23 showed 70%–95% similarity when queried with phage databases on both tax_myPHAGE and PhaBOX2 tools. Therefore, our isolate represents a new species of the genus *Drulisvirus*, which we have named *Drulisvirus cayetanensis*.

Further assessment using an *in vivo* model was necessary to evaluate the safety and efficacy in a preclinical setting (10). The *G. mellonella* model was selected because it possesses an immune system mediated by phagocytes (hemocytes) and soluble molecules composed of complement-like proteins, anti-microbial peptides, and melanin, which immobilize or kill pathogens (19, 54). Critically, when compared with *D. melanogaster*, *G. mellonella* can withstand an incubation temperature of 37°C, making it a suitable model for investigating Gram-negative human pathogens (20).

Our findings showed that GA23 led to a significant increase in larva survival when treated 1 h after the bacterial inoculum, resulting in 100% viability at the 96 h endpoint. Although we did not assess the effects of earlier phage inoculation, this result provides further support to the argument that a short time between viral and bacterial inoculation leads to improved survival (44). No mortality was observed in the groups treated with SM or viral lysate, ruling out the possibility of traumatic inoculation injuries, chemical intoxication from the SM buffer components, and viral lysate endotoxins (11).

A key observation from this study is that the efficiency of the *in vivo* model can be optimized by appropriately selecting multiplicity of infection, as higher values do not necessarily correlate with greater mortality reduction *in vivo* (36). At the optimal MOI, GA23 demonstrates superior efficiency compared to other phage isolates using the *G. mellonella* model. Specifically, GA23 outperforms SRD2021, which results in 60% larval survival at 24 h; SXFY507, which achieves 60% survival at 72 h; and BUCT610, which results in 83% survival at 72 h (13, 37, 44). These findings—generated using a robust invertebrate model as an alternative to murine models—highlight the therapeutic potential of GA23 for treating systemic *K. pneumoniae* infections (20, 55).

## Conclusion

A new bacteriophage species, named *Drulisvirus cayetanensis*, was isolated and characterized, demonstrating 100% protection against lethal multidrug-resistant *K. pneumoniae* infection. Despite its limited host range, the phage exhibited strong lytic activity, supported by depolymerase enzymes, a high virion production rate, and stability across a broad range of temperatures and pH levels. WGS confirmed the absence of lysogeny-associated genes, antibiotic resistance genes, and virulence genes, eliminating the risk of genetic transfer. The *Galleria mellonella* model provided *in vivo* pre-clinical evidence of the phage's therapeutic potential, highlighting the novel phage's superiority to other reported phages in combating *K. pneumoniae* infection.

## ACKNOWLEDGMENTS

The authors acknowledge the help of Maricielo Orellano for supplying the *Galleria mellonella* larvae. They also acknowledge Dr. David Bacsik for his careful and detailed review of this manuscript.

This research received no external funding.

G.Q-.V.: Conceptualization (lead), Data curation (lead), Formal analysis (lead), Investigation (lead), Methodology (equal), Project administration (equal), Validation (supporting), Writing – original draft (equal), Writing-Review and Editing (equal). G.I.A-.L.: Conceptualization (supporting), Investigation (supporting), Methodology (equal), Project administration (equal), Validation (supporting), Visualization (lead), Writing – original draft (equal). J.T.: Conceptualization (supporting), Methodology (equal), Project administration (equal), Resources (lead), Supervision (lead), Validation (lead), Writing-Review and Editing (equal). D.C.: Investigation (supporting), Methodology (equal), Writing- Review and Editing (equal), Data curation, Software (equal). R.L.: Investigation (supporting), Data curation (supporting), Software (equal). B.A.: Investigation (supporting), Resources (supporting), Data curation (supporting), Software (equal). P.T.: Investigation (supporting), Writing-Review and Editing (equal), Resources (supporting).

## AUTHOR AFFILIATIONS

[1]Laboratorio de Resistencia Antibiótica y Fagoterapia, Laboratorios de Investigación y Desarrollo, Facultad de Ciencias e Ingeniería, Universidad Peruana Cayetano Heredia, Lima, Peru

[2]Facultad de Medicina, Universidad Peruana Cayetano Heredia, Lima, Peru

[3]Laboratorio de Genómica Microbiana, Laboratorios de Investigación y Desarrollo, Facultad de Ciencias e Ingeniería, Universidad Peruana Cayetano Heredia, Lima, Peru

[4]Laboratorio de Moléculas Individuales, Laboratorios de Investigación y Desarrollo, Facultad de Ciencias e Ingeniería, Universidad Peruana Cayetano Heredia, Lima, Peru

[5]Instituto de Medicina Tropical Alexander von Humboldt, Universidad Peruana Cayetano Heredia, Lima, Peru

[6]Parasites and Microbes Programme, Wellcome Sanger Institute, Hinxton, United Kingdom

## AUTHOR ORCIDs

Gustavo Quispe-Villegas  http://orcid.org/0000-0001-8833-5072
Pablo Tsukayama  http://orcid.org/0000-0002-1669-2553
Jesús Tamariz  http://orcid.org/0000-0002-0827-8117

## AUTHOR CONTRIBUTIONS

Gustavo Quispe-Villegas, Conceptualization, Data curation, Formal analysis, Investigation, Methodology, Project administration, Validation, Writing – original draft, Writing – review and editing | Gabriela I. Alcántara-Lozano, Conceptualization, Investigation, Methodology, Project administration, Validation, Visualization, Writing – original draft | Diego Cuicapuza, Data curation, Investigation, Methodology, Software, Writing – review and editing | Brenda Ayzanoa, Data curation, Investigation, Resources, Software | Pablo Tsukayama, Investigation, Resources, Writing – review and editing | Jesús Tamariz, Conceptualization, Investigation, Methodology, Project administration, Resources, Supervision, Validation, Writing – review and editing.

## DATA AVAILABILITY

The raw read files and assemblies for the isolate are available at NCBI under BioProject accession number PRJNA1192385.

## ADDITIONAL FILES

The following material is available online.

Open Peer Review

**PEER REVIEW HISTORY (review-history.pdf).** An accounting of the reviewer comments and feedback.

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
