## [Reviewer comments · Microbiology Spectrum]

Microbiology Spectrum

***In vivo* evaluation of phage therapy against *Klebsiella pneumoniae* using the *Galleria mellonella* model and molecular characterization of a novel *Drulisvirus* phage species**

Gustavo Quispe-Villegas, Gabriela Alcántara-Lozano, Diego Cuicapuza, Raúl Laureano Revilla, Brenda Ayzanoa, Pablo Tsukayama, and Jesus Tamariz

Corresponding Author(s): Jesus Tamariz, Universidad Peruana Cayetano Heredia

Review Timeline:

Submission Date:	May 7, 2024
Editorial Decision:	June 20, 2024
Revision Received:	December 10, 2024
Editorial Decision:	January 27, 2025
Revision Received:	February 24, 2025
Accepted:	March 7, 2025

Editor: Bobby Warren

Reviewer(s): Disclosure of reviewer identity is with reference to reviewer comments included in decision letter(s). The following individuals involved in review of your submission have agreed to reveal their identity: Ayaz Ahmed (Reviewer #1); Nana Ama Amissah (Reviewer #2)

Transaction Report:

DOI: <https://doi.org/10.1128/spectrum.01145-24>

Re: Spectrum01145-24 (*In-vivo* Efficacy of Phage Therapy Against *Klebsiella pneumoniae* Using a *Galleria mellonella* Model)

Dear Dr. Gustavo Alonso Quispe-Villegas:

Thank you for the privilege of reviewing your work. Below you will find my comments, instructions from the Spectrum editorial office, and the reviewer comments.

Revision Guidelines

Sincerely,
Bobby Warren
Editor
Microbiology Spectrum

Reviewer #1 (Comments for the Author):

I would suggest authors revise the manuscript and improve their writing.

Results and discussion, which are the essence of the paper, are poorly written, and no proper analysis was done.

The abstract needs to be rewritten, especially the conclusion.

In the plaque isolation heading, why was plaque picked up in BHI medium but not in phage buffer? What is the reason behind the 1-hour incubation of phage plaque in BHI? Doing so, what author wanted to check?

What was the control for pH and thermal stability? Statistics need to be done in the pH and thermal stability experiment by selecting any control temperature and pH value. Additionally, statistical tests can be applied to determine significant changes in titer with respect to control.

Phage pH stability could be evaluated at a higher alkaline pH (9-14) as well.

Thermal stability was measured at a 1-degree change. It is obvious that a change in titer cannot be observed; the temperature range can be increased by at least 5 degrees.

Eclipse (if chloroform was added to the aliquot at each interval) and latent period can be found by a one-step growth curve, which is not calculated. Kindly add a latent period for phage.

For phage therapy, phenotypic and genotypic characterization is necessary, which determines the phage family and safety on a genotypic level. Why were these experiments not performed?

In Fig. 1, the morphology of phage GA23 seems to be not a clonal plaque. There are different plaques with variable sizes. A smaller halo plaque is different from other plaques. Kindly update the manuscript with a plaque plate that shows uniform morphology.

In-vivo Efficacy of Phage Therapy Against *Klebsiella pneumoniae* Using a *Galleria mellonella* Model

Gustavo A. Quispe-Villegas^{1,2}, Gabriela I. Alcántara-Lozano^{1,2}, Jesús Tamariz^{1,2}

¹Laboratorio de Resistencia Antibiótica y Fagoterapia, Universidad Peruana Cayetano Heredia, Lima, Peru

²Facultad de Medicina, Universidad Peruana Cayetano Heredia, Lima, Peru

Abstract

Introduction: Multidrug resistant (MDR) *K. pneumoniae* represents a threat to healthcare systems and continues to pose a challenge to conventional antibiotic therapy. Phage therapy represents an alternative treatment that can be evaluated using novel *in-vivo* infection models, such as the *Galleria mellonella* model, allowing for optimized efficacy assessment.

Methods: We used an ESBL-producing *K. pneumoniae* isolate to identify lytic bacteriophages from hospital wastewater. Phage stability at different temperatures and pH values, host range assays, optimal *in-vitro* multiplicity of infection (MOI), one-step growth curve and the *Galleria mellonella* infection model were performed.

Results: The isolated phage GA23 demonstrated a narrow host range and lytic activity across different temperatures and pH ranges. GA23 significantly improved the survival of *Galleria mellonella* larvae after 96 hours.

Conclusions: The *Galleria mellonella* model is an alternative that allows for the evaluation of therapeutic efficacy against MDR *K. pneumoniae* without using murine models.

Keywords: Bacteriophages, phage therapy, *Klebsiella pneumoniae*, *Galleria mellonella*

Importance

The use of a non-murine model such as *Galleria mellonella* facilitates the *in vivo* study of novel antimicrobial agents like phage therapy, which represents an alternative to antibiotics in combating antimicrobial resistance. This model can be

conducted in less time, requires lower logistical demands, and yields comparable significant results to traditional rodent models.

Background

Klebsiella pneumoniae is a gram-negative bacterium of the *Enterobacterales* order and is recognized as one of the most important opportunistic pathogens; responsible for nosocomial infections, including soft tissue infections, urinary tract infections, intra-abdominal infections, pneumonia, bacteremia, septicemia, and septic shock¹⁻³. Due to hospital recurrence, the use of antibiotics over several decades has resulted in the emergence of multidrug-resistant (MDR) strains of *K. pneumoniae*⁴⁻⁵. Infections caused by MDR *K. pneumoniae* contribute to increased mortality, worse clinical outcomes, prolonged hospital stays, inappropriate overuse of antibiotics, and a perpetual threat to healthcare systems^{2,5-6}. It is estimated that in 2019, 4.95 million deaths associated with resistant bacteria occurred worldwide, with *K. pneumoniae* included in the list of pathogens⁷. Additionally, according to the World Bank, antimicrobial resistance is projected to cost up to 6.1 billion dollars annually by 2050 worldwide⁸.

Because of the morbimortality of *K. pneumoniae* MDR, promising therapeutics, such as phage therapy, have regained attention as they represent an alternative treatment for infections caused by antibiotic-resistant bacteria⁹⁻¹². Several studies have reported the application of phage therapy in severe infections, such as in curative treatment for patients with recurrent urinary tract infections caused by MDR *K. pneumoniae*¹³⁻¹⁴. However, not all phages are suitable for therapeutic purposes; thus, identification and characterization are fundamental initial stages in any phage-related research^{12,15}. Those with lytic replication cycles are preferred because of a lower risk of horizontal gene transfer by transduction to pathogens, which may include virulence factors¹⁵⁻¹⁶.

Furthermore, the evaluation of *in-vivo* efficacy has been improved with the introduction of the *Galleria mellonella* model, an insect of the order *Lepidoptera* and the family *Pyralidae*, which possesses a series of characteristics in its larval stage, making it a suitable model: it is economical, easy to maintain, and does not require specialized laboratory equipment¹⁷⁻¹⁹. Additionally, it represents an alternative to murine models for the study of microbial infections, without the

disadvantage of high maintenance costs and reproduction time. The ethical scope and the lesser complexity of *Galleria spp.* make it an alternative model for *in-vivo* assessment of virulence of gram-negative bacteria¹⁹⁻²⁰. Based on the above, the present study was conducted with the aim of applying phage therapy in the *G. mellonella in-vivo* model against an MDR *K. pneumoniae* isolate.

Materials and Methods

Bacterial isolates

The isolate of *Klebsiella pneumoniae* utilized in this study was an extended-spectrum beta-lactamase (ESBL)-producing strain, obtained from the strain collection of the Laboratorio de Resistencia a Antimicrobianos y Fagoterapia. Seven more carbapenemase-producing *K. pneumoniae* isolates were also obtained for the host range assay. All isolates were cryopreserved in Tryptic Soy Broth (TSB) with 20% glycerol at -20°C. All bacterial cultures for experimentation were performed on Luria Bertani (LB) agar (LB with 1.5% weight/volume agar) at 37° for 24 hours. The aforementioned ESBL strain was used in the isolation and characterization of the phage, as well as in *in-vivo* experiments with the *Galleria mellonella* model; while the carbapenemase-producing *K. pneumoniae* were only used in the host range assay.

Phage isolation

Phages were isolated from wastewater from the drainage of a tertiary hospital in Lima, Peru. A sample of 10 mL was obtained and subsequently subjected to filtration using 0.22 µm pore-size syringe filters to remove debris and bacteria. This procedure was repeated three times. For enrichment, 200 µL of bacterial solution in 0.9% normal saline with a turbidity of 0.5 McFarland units was inoculated into 1 mL of filtered wastewater in 90 ml of double-concentration Luria Bertani (LB) broth, which was then incubated overnight at 37°C in a shaker at 110 rpm for 24 hours. Subsequently, three filtrations were performed using 0.22 µm pore-size syringe filters to remove cells. This filtrate was serially diluted to 10⁻⁴ and 10⁻⁶ in 0.9% normal saline.

We used an overlay agar plaque assay: for bottom agar plates, LB broth with 1.5% weight/volume agar was used. A mix of 100 µL of bacterial solution 0.9%

normal saline with a turbidity of 0.5 McFarland units was inoculated with 100 μ L of either diluted viral filtrate, incubated at room temperature for 5 minutes and mixed with 5 mL of molten top LB-semisolid agar (LB with 0.4% weight/volume agar). The mix was poured onto LB agar plates and plates were allowed to set. Overnight incubation was performed on inverted plates at 37°C and the next day the plates were examined to identify clear plaques.

Single plaques with clear lytic halos were picked with 1000 μ L sterile pipette tips and incubated with 1.5 mL of brain-heart infusion (BHI) medium. The solution was incubated at 37°C for 1 hour, followed by centrifugation for 2 minutes at 1500 rpm. The supernatant was collected, and the filtrate was subjected to three subsequent rounds of plaque assay for the isolation and expansion of clonal phages. Purified phages were then stored at 8°C.

Phage-host range assay

Aside from the isolation bacterial strain, which was used to isolate phages, the host range was determined against six different clinical carbapenemase-producing *K. pneumoniae* isolates (VIM-1, IMP-8, NDM-1, KPC-2, OXA-48-like and OXA-181) by standard spot assays. Aliquots of 2 μ l diluted phage lysate (10^8 PFU/ml) were spotted on bacterial lawns over Luria Bertani (LB) agar. Bacterial cultures were suspended on 0.9% normal saline and titrated to 0.5 McFarland scale. After incubating for 37°C for 24 hours, the spots were assessed by the formation of lytic plaques without turbidity for positive infection. The experiment was performed as independent duplicates.

Phage stability at different temperature and pH ranges

For pH testing, SM buffer (50mM Tris-HCl, 8mM magnesium sulfate, 100mM sodium chloride, 0.01% gelatin, pH 7.5) was prepared and adjusted to pH values of 4, 5, 6, 7, and 8. In sterile test tubes, a viral solution of 10^8 PFU/mL was added in a 1:10 ratio with SM buffer. The mixture was incubated at 37°C in a shaker at 210 rpm for 1 hour. For temperature stability, a viral solution in 0.9% normal saline was added in a 1:10 ratio, followed by incubation at 8, 35, 36, 37, 38, 39, 40, and 50°C for 1 hour. For both experiments, lytic activity was evaluated with the aforementioned agar plaque assay.

Optimal multiplicity of infection (MOI) and one-step grown curve

The multiplicity of infection (MOI) is the ratio between the plaque forming units (PFU/mL) and the colony forming units (CFU/mL). A bacterial solution in 0.9% normal saline was obtained from a log-phase culture of the bacterial isolate (OD₆₀₀ = 0.6), and was mixed with viral filtrate to obtain a MOI of 0.1, 1 and 10. The mixture was cultivated at 37°C for 4 hours and phages were filtered with 0.22 µm pore-size syringe filters. An agar plaque assay was performed in duplicate, and the optimal MOI was considered to be the one that presented the highest PFU.

For the one-step grown curve, a bacterial-phage mixture in 0.9% normal saline at the optimal MOI was obtained as mentioned above. Subsequently, the **mix** was incubated for 15 minutes at 37°C in a shaker at 100 rpm, followed by centrifugation at 3000 rpm in 2 cycles of 10 minutes each. The supernatant was separated, and the pellet was resuspended in 1 mL of LB broth. **Then, 100 µl of the obtained solution was added to 15 mL of LB broth tempered at 37°C. This was considered as time zero, from which duplicate samples of 1 mL of broth were taken every 10 minutes for a total duration of 110 minutes.** These samples were utilized for agar plaque assays to determine viral concentration via serial dilutions spanning from 10⁻² to 10⁻⁸.

Galleria mellonella in-vivo infection model

Larvae of *G. mellonella* without dark spots and with a length greater than 2.5 cm were selected. Decontamination was performed using a sterile cotton swab soaked in 70% ethanol. All bacterial inoculations were carried out using sterile 300 µL insulin syringes into the last left posterior proleg. Viral inoculations were administered into the last right posterior proleg. The larvae were then incubated in Petri dishes at 35°C in darkness and were assessed every 24 hours for a total of 96 hours. Larvae were considered dead if there was a loss of spontaneous movement without response to tactile stimulation using sterile 100 µL pipette tips.

A bacterial solution in 0.9% saline (OD₆₀₀ = 0.6) was prepared and diluted to 10⁷ CFU/mL with SM buffer. Concomitantly, the viral lysate was diluted with buffer SM until reaching the concentration needed to match the optimal MOI.

A cohort of 20 randomly chosen larvae was distributed among 4 groups of 5 larvae each: a control group was inoculated with 200 µL of SM buffer; a second group received an inoculum of 200 µL of bacterial suspension; a third group received 200 µL of viral concentrate; and a fourth group was inoculated with a mixture of bacterial and viral suspensions in a 1:1 ratio (200 µL each). For the last group, the bacterial suspension was initially inoculated, and after 60 minutes, the bacteriophage suspension was added. Experiments were performed independently in triplicate and data were pooled.

Statistics

With the larval survival results over the 96-hour period, a Kaplan-Meier curve was constructed, and the log-rank test was performed to compare survival curves. P values <0.05 were considered statistically significant. All statistical analyses and graphics were conducted using R-Studio software v4.2.2 and the statistical packages ggplot2 v3.3.4, survival v3.5-7, and dplyr v1.1.4.

Results

Isolation of bacteriophage GA23

The wastewater incubated with the ESBL-producing *K. pneumoniae* isolate has successfully yielded multiple bacteriophages capable of forming lytic halos, as demonstrated by the agar plaque assay method. Three phages with the biggest diameters (9 mm) were selected. However, subsequent passages for enrichment and isolation of a clonal virus resulted only in one phage with enough titers ($>10^8$ PFU/ml) for experimentation. This bacteriophage was called GA23. Around the central plaque, the formation of a secondary semi-transparent halo was observed (Fig. 1).

Host range profile of GA23

When tested against seven different carbapenemase-producing *K. pneumoniae*, GA23 exhibited null activity and failed to infect either strain aside from the ESBL-producing *K. pneumoniae* (Table 1).

pH and temperature stability of GA23

The bacteriophage GA23 preserved its lytic activity from pH 4 to pH 8 (Fig. 2). No tenfold titer change was observed. Regarding thermal stability, a maximum titer was obtained at 8°C, and remaining constant between 35-40°C. However, no phages survived when the temperature reached 50°C (Fig. 2).

Optimal MOI and one-step growth curve of GA23

When the MOI was 0.1, the titers of phage GA23 reached their maximum value. One-step growth curve and the *in-vivo* assay with *G. mellonella* experiments were conducted at this concentration. The one-step growth curve indicated that the exponential phase lasted approximately 30 minutes, with a burst size of 161 viral particles/cell (Fig. 3).

Galleria mellonella in-vivo infection model

In the *G. mellonella* model, the viability of the control group inoculated with bacterial solution was 53% on day 1, dropping to 26.7% by day 4. No mortality was observed among the groups treated with SM buffer or viral lysate. When inoculating phage GA23 60 minutes after the injection of the bacterial inoculum at a MOI of 0.1, the viability of up to 100% of the larvae was preserved at 96 hours. Statistical analysis using the log-rank test revealed a significant difference in survival curves between the untreated control and the treated larvae ($p=0.007$) (Fig. 4).

Discussion

The main objective of our study was to evaluate phage therapy against a MDR isolate of *K. pneumoniae* and validate the *Galleria mellonella* model for phage therapy against this pathogen. The narrow host range exhibited by GA23 could be explained by the specificity of the receptor-binding proteins (RBPs) of the phage and the reversible interaction with the capsule and K membrane antigen of *Klebsiella spp*²¹. Due to the critical role of membrane components in bacterial susceptibility²²⁻²³, preventing adsorption constitutes the main mechanism of resistance to bacteriophages²⁴, which is often achieved through capsular and membrane mutations. Among the latter, insensitivity to the adsorption of phage GH-K3 is exemplified, dependent on the gene activity of OmpC (and porin OmpK36), GT1, GT2, and wcaJ²⁴⁻²⁶; as well as the NPat and BMac

bacteriophages, impermeable to acapsular strains of *K. pneumoniae* due to alterations in the *wzb* and *wzc* genes²⁶.

Another factor involved in the bacteriophage-host interaction is the presence of viral depolymerases in RBPs²⁵. These enzymes have the property of being released as free enzymes, which retain catalytic activity towards the capsule²². The latter is responsible for the macroscopic phenomenon observed in phage GA23, and in phages possessing depolymerases: the macroscopic creation of a halo surrounding the plaque, resulting from the denudation of the CPS^{21-22,27-28}. All these factors limit the application of a single bacteriophage towards multiple strains of the *Klebsiella* genus²⁹⁻³⁰, which is crucial when covering clinical strains belonging to multiple K genotypes²⁸, where a single broad-spectrum bacteriophage is insufficient for all described capsular types³¹.

It should be noted that, in addition to the biological limitations for developing broad-spectrum virions with lytic activity, the microenvironment conditions that allow phage proliferation play a role enabling the success of the model both *in-vitro* and *in-vivo*, in the medical and industrial applicability of bacteriophages³². Bacteriophages retaining lytic activity have been reported from temperatures as low as -20°C³³⁻³⁴ to as high as 60-70°C^{13,35-37}, which is relevant for phage storage through refrigeration, and their subsequent applicability towards endothermic or ectothermic *in-vivo* models, with the latter being the case of *G. mellonella*³⁸, employed in our study. Phage GA23 maintains viral particle formation at temperatures ranging from 8°C to 40°C, a range that would not limit its application in the treatment of infections in warm-blooded humans and mammals.

In parallel, the pH of the microenvironment becomes relevant in *in-vivo* models when considering the different compartments of an organism and the routes of administration of bacteriophages³⁹: human skin has an alkaline pH secondary to bacterial colonization, while the renal parenchyma and the urinary tract have drastic acid-base environments under both physiological and morbid conditions³². Phage GA23 shows stability in the pH range of 4-8, whereas in other studies, phages such as IME268 have been reported to preserve their biological activity at pH from 2 to 14⁴⁰. Other reported bacteriophages retaining functionality at pH 4-12 include Kp vB_Kpn_ZCKp20p³⁴, vB_KpnP_k3_ULINTkp¹³⁶, BUCT610¹³, and vB_KpnS_SXFY507³⁷, while vB_KpnP_Klvazma has the limitation of

generating lytic viral particles at pH 4-5⁴¹. Following the examples provided, there is difficulty in administering phage therapy orally, given that the gastric pH reaches up to 1³².

Regarding the multiplicity of infection (MOI), its relevance lies in optimizing the generation of new virions during each lytic cycle. This value varies among viral strains and is influenced by the size of the virion, bacterial size, receptor availability, and, in parallel, bacterial cell division presenting new hosts against the increasing viral load³⁵. Theoretically, a higher MOI leads to a greater reduction in bacterial concentration³³; however, phages optimized at an MOI of 1000³⁶ as well as at 0.001^{32,40} have been described. *In-vivo* assays conducted against *G. mellonella* should be performed at an optimal MOI, with a direct influence on the survival rate of the model: for example, at an MOI of 100, vB_KpnP_k3_ULINTkp1 rescues 53% of the larvae in 72 hours, compared to vB_KpnP_k3_ULINTkp2 which only achieves a survival rate of 27%³⁶. In this study, the optimal MOI was found to be 0.1, this finding allowed the development of the survival assay theoretically ensuring the highest production of virions.

The one-step growth curve has allowed the identification of the latent period and burst size, with the latter described as the viral particles generated from the lysis of a bacterium at the end of the logarithmic phase. Phage GA23 begins to generate virions at 5 minutes into its cycle and can produce up to 161 phages per bacterium. Comparatively, latent periods can last between 5 minutes³⁵⁻³⁶ and one hour⁴², while burst size can reach up to 650 CFU/ml per bacterium³³. A shorter latent period and larger burst size lead to a higher probability of viral proliferation, but along with MOI, thermal and acid-base stability, these kinetic characteristics are unique to each bacteriophage^{35,42}.

On the other hand, an *in-vivo* model relies on viral kinetics and allows for the measurement of bacteriophage efficacy as therapy, in addition to serving as a safety assay in a preclinical setting¹⁰. Phage GA23 has achieved a drastic increase in survival in *G. mellonella* when inoculated after 60 minutes of bacterial inoculum, resulting in 100% larval viability at 96 hours. This further adds to the argument that the shorter the time between viral and bacterial inoculation, the greater the survival of the model⁴³. No mortality was evident among the groups treated with SM buffer or viral lysate, ruling out the possibility of mortality

attributable to traumatic injuries from inoculation or chemical intoxication by the components of the buffer and viral lysate.

It is noteworthy that the efficiency of the *in-vivo* model can be optimized through MOI, yielding different survival outcomes. It should be calculated for each phage, without assuming that a higher MOI results in a greater reduction in mortality *in vivo*³⁶. At optimal MOI, when compared with different phage isolates, the efficiency of GA23 stands out in comparison to SRD2021, which allows for 60% larval survival at 24 hours; SXFY507, with 60% survival at 72 hours, and the mentioned BUCT610, up to 83% at 72 hours^{13,37,43}. These results suggest that therapeutic applicability that can be tested without the need for murine models, to subsequently be exploited alongside antibiotic therapy in the treatment of infectious morbidities in future preclinical trials²¹.

Conclusion

Phage therapy retains therapeutic applicability that can be tested in MDR *K. pneumoniae* without the need for murine models. Likewise, through survival experiments in *G. mellonella* against *K. pneumoniae* infection, it has validated the *in-vivo* application of bacteriophage GA23.

Ethical statements

According to our Institutional Review Board, ethical authorization was not necessary for the utilization of the bacterial isolates, as well as for the *Galleria mellonella* larvae. The project was approved under the Dirección Universitaria de Asuntos Regulatorios de la Investigación resolution CAR-DUARI-205-23.

Acknowledgments

The authors acknowledge the help of Diego Cuicapuza for providing the *Klebsiella pneumoniae* isolate, and of Maricielo Orellano for supplying the *Galleria mellonella* larvae. They also acknowledge the Laboratorio de Resistencia Antibiótica y Fagoterapia, hosted at the Universidad Peruana Cayetano Heredia, for the logistical support in conducting the experiments.

Author's Contributions

Gustavo A. Quispe-Villegas: Conceptualization (lead), Data curation (lead), Formal analysis (lead), Investigation (lead), Methodology (equal), Project administration (equal), Validation (supporting), Writing – original draft (equal)

Gabriela I. Alcántara-Lozano: Conceptualization (supporting), Investigation (supporting), Methodology (equal), Project administration (equal), Validation (supporting), Visualization (lead), Writing – original draft (equal)

Jesús Tamariz: Conceptualization (supporting), Methodology (equal), Project administration (equal), Resources (lead), Supervision (lead), Validation (lead)

Author Disclosure Statement

No competing financial interests exist.

Funding Information

This research received no external funding.

References

1. Li M, Li P, Chen L, Guo G, Xiao Y, Chen L, et al. Identification of a phage-derived depolymerase specific for KL64 capsule of *Klebsiella pneumoniae* and its anti-biofilm effect. *Virus Genes*. 2021 Oct 1;57(5):434–42. doi: 10.1007/s11262-021-01847-8.
2. Feng J, Gao L, Li L, Zhang Z, Wu C, Li F, et al. Characterization and genome analysis of novel *Klebsiella* phage BUCT556A with lytic activity against carbapenemase-producing *Klebsiella pneumoniae*. *Virus Res*. 2021 Oct 2;303. doi: 10.1016/j.virusres.2021.198506.
3. Wintachai P, Naknaen A, Thammaphet J, Pomwised R, Phaonakrop N, Roytrakul S, et al. Characterization of extended-spectrum- β -lactamase producing *Klebsiella pneumoniae* phage KP1801 and evaluation of therapeutic efficacy in vitro and in vivo. *Sci Rep*. 2020 Dec 1;10(1). doi: 10.1038/s41598-020-68702-y.
4. Venturini C, Ben Zakour NL, Bowring B, Morales S, Cole R, Kovach Z, et al. Fine capsule variation affects bacteriophage susceptibility in *Klebsiella pneumoniae* ST258. *FASEB Journal*. 2020 Aug 1;34(8):10801–17. doi: 10.1096/fj.201902735R.
5. Li N, Zeng Y, Bao R, Zhu T, Tan D, Hu B. Isolation and Characterization of Novel Phages Targeting Pathogenic *Klebsiella pneumoniae*. *Front Cell Infect Microbiol*. 2021 Dec 3;11. doi: 10.3389/fcimb.2021.792305.
6. Walther-Rasmussen J, Høiby N. Class A carbapenemases. Vol. 60, *J. Antimicrob. Chemother*. 2007. p. 470–82. doi: 10.1093/jac/dkm226.
7. Murray CJ, Ikuta KS, Sharara F, Swetschinski L, Robles G, Gray A, et al. Global burden of bacterial antimicrobial resistance in 2019: a systematic analysis. *Lancet*. 2022 Feb 12;399(10325):629–55. doi: 10.1016/S0140-6736(21)02724-0.

8. Final Report. Drug Resistant Infections: A Threat to Our Economic Future [Internet]. March 2017. Available from: www.worldbank.org [Last accessed: 01/02/2024]
9. Corbellino M, Kieffer N, Kutateladze M, Balarjishvili N, Leshkasheli L, Askilashvili L, et al. Eradication of a multi-drug resistant, carbapenemase-producing *Klebsiella pneumoniae* isolate following oral and intra-rectal therapy with a custom-made, lytic bacteriophage preparation. *Clin Infect Dis*. 2020 Apr 15;70(9):1998-2001. doi: 10.1093/cid/ciz782.
10. Roach DR, Debarbieux L. Phage therapy: Awakening a sleeping giant. *Emerg Top Life Sci*. 2017 Apr 21;1(1):93-103. doi: 10.1042/ETLS20170002.
11. Brown R, Lengeling A, Wang B. Phage engineering: how advances in molecular biology and synthetic biology are being utilized to enhance the therapeutic potential of bacteriophages. *Quant Biol* 5, 42–54 (2017). doi: 10.1007/s40484-017-0094-5.
12. Prada-Peñaranda C, Holguin-Moreno A, González-Barrios A, Vives-Florez M. Fagoterapia, alternativa para el control de las infecciones bacterianas. *Perspectivas en Colombia*. Universitas Scientiarum. 2014 Aug. 14;20(1):43-60. doi: 10.11144/Javeriana.SC20-1.faci
13. Pu M, Han P, Zhang G, Liu Y, Li Y, Li F, et al. Characterization and Comparative Genomics Analysis of a New Bacteriophage BUCT610 against *Klebsiella pneumoniae* and Efficacy Assessment in *Galleria mellonella* Larvae. *Int J Mol Sci*. 2022 Jul 1;23(14). doi: 10.3390/ijms23148040.
14. Bao J, Wu N, Zeng Y, Chen L, Li L, Yang L, et al. Non-active antibiotic and bacteriophage synergism to successfully treat recurrent urinary tract infection caused by extensively drug-resistant *Klebsiella pneumoniae*. *Emerg Microbes Infect*. 2020 Jan 1;9(1):771–4. doi: 10.1080/22221751.2020.1747950.
15. Luong T, Salabarria AC, Roach DR. Phage Therapy in the Resistance Era: Where Do We Stand and Where Are We Going? *Clin Ther*. 2020 Sep 1;42(9):1659–80. doi: 10.1016/j.clinthera.2020.07.014.
16. Hibstu Z, Belew H, Akelew Y, Mengist H. Phage Therapy: A Different Approach to Fight Bacterial Infections. *Biologics*. 2022 Oct 6;16:173–86. doi: 10.2147/BTT.S381237.
17. Nath S, Moussavi F, Abraham D, Landman D, Quale J. In vitro and in vivo activity of single and dual antimicrobial agents against KPC-producing *Klebsiella pneumoniae*. *J Antimicrob Chemother*. 2018 Feb 1;73(2):431-436. doi: 10.1093/jac/dkx419.
18. Göttig S, Frank D, Mungo E, Nolte A, Hogardt M, Besier S, et al. Emergence of ceftazidime/avibactam resistance in KPC-3-producing *Klebsiella pneumoniae* in vivo. *J Antimicrob Chemother*. 2019 Nov 1;74(11):3211-3216. doi: 10.1093/jac/dkx419.
19. Tsai CJY, Loh JMS, Proft T. *Galleria mellonella* infection models for the study of bacterial diseases and for antimicrobial drug testing. *Virulence*. 2016 Apr 2;7(3):214-29. doi: 10.1080/21505594.2015.1135289.
20. Ménard G, Rouillon A, Cattoir V, Donnio P. *Galleria mellonella* as a Suitable Model of Bacterial Infection: Past, Present and Future. *Front Cell Infect Microbiol*. 2021 Dec 22;11:782733. doi: 10.3389/fcimb.2021.782733.

21. Labrie SJ, Samson JE, Moineau S. Bacteriophage resistance mechanisms. *Nat Rev Microbiol.* 2010 May;8(5):317-27. doi: 10.1038/nrmicro2315.
22. Knecht LE, Veljkovic M, Fieseler L. Diversity and Function of Phage Encoded Depolymerases. *Front Microbiol.* 2020 Jan 10;10:2949. doi: 10.3389/fmicb.2019.02949.
23. Squeglia F, Maciejewska B, Latka A, Ruggiero A, Briers Y, Drulis-Kawa Z, et al. Structural and Functional Studies of a *Klebsiella* Phage Capsule Depolymerase Tailspike: Mechanistic Insights into Capsular Degradation. *Structure.* 2020 Jun 2;28(6):613-624.e4. doi: 10.1016/j.str.2020.04.015
24. Cai R, Wang G, Le S, Wu M, Cheng M, Guo Z, et al. Three capsular polysaccharide synthesis-related glucosyltransferases, GT-1, GT-2 and WcaJ, are associated with virulence and phage sensitivity of *Klebsiella pneumoniae*. *Front Microbiol.* 2019 May 28;10:1189. doi: 10.3389/fmicb.2019.01189.
25. Cai R, Wu M, Zhang H, Zhang Y, Cheng M, Guo Z, et al. A smooth-type, phage-resistant *Klebsiella pneumoniae* mutant strain reveals that OmpC is indispensable for infection by phage GH-K3. *Appl Environ Microbiol.* 2018 Nov 1;84(21). doi: 10.1128/AEM.01585-18.
26. Dunstan RA, Bamert RS, Tan KS, Imbulgoda U, Barlow CK, Taiaroa G, et al. Epitopes in the capsular polysaccharide and the porin OmpK36 receptors are required for bacteriophage infection of *Klebsiella pneumoniae*. *Cell Rep.* 2023 Jun 27;42(6). doi: 10.1016/j.celrep.2023.112551.
27. Fang Q, Zong Z. Lytic Phages against ST11 K47 Carbapenem-Resistant *Klebsiella pneumoniae* and the Corresponding Phage Resistance Mechanisms. *mSphere.* 2022 Apr 27;7(2):e0008022. doi: 10.1128/msphere.00080-22.
28. Concha-Eloko R, Barberán-Martínez P, Sanjuán R, Domingo-Calap P. Broad-range capsule-dependent lytic *Sugarlandvirus* against *Klebsiella* sp. *Microbiol Spectr.* 2023 Dec 12;11(6):e0429822. doi: 10.1128/spectrum.04298-22.
29. Egado JE, Costa AR, Aparicio-Maldonado C, Haas PJ, Brouns SJJ. Mechanisms and clinical importance of bacteriophage resistance. *FEMS Microbiol Rev.* 2022 Feb 9;46(1):fuab048. doi: 10.1093/femsre/fuab048.
30. Dunstan RA, Bamert RS, Belousoff MJ, Short FL, Barlow CK, Pickard DJ, et al. Mechanistic Insights into the Capsule-Targeting Depolymerase from a *Klebsiella pneumoniae* Bacteriophage. *Microbiol Spectr.* 2021 Sep 3;9(1). doi: 10.1128/Spectrum.01023-21.
31. Lourenço M, Osbelt L, Passet V, Gravey F, Megrian D, Strowig T, et al. Phages against Noncapsulated *Klebsiella pneumoniae*: Broader Host range, Slower Resistance. *Microbiol Spectr.* 2023 Aug 17;11(4). doi: 10.1128/spectrum.04812-22.
32. Bai J, Zhang F, Liang S, Chen Q, Wang W, Wang Y, et al. Isolation and Characterization of vB_kpnM_17-11, a Novel Phage Efficient Against Carbapenem-Resistant *Klebsiella pneumoniae*. *Front Cell Infect Microbiol.* 2022 Jul 5;12. doi: 10.3389/fcimb.2022.897531.
33. Faye MS, Hakim TA, Zaki BM, Makky S, Abdelmoteleb M, Essam K, et al. Morphological, biological, and genomic characterization of *Klebsiella pneumoniae* phage vB_Kpn_ZC2. *Virol J.* 2023 Dec 1;20(1). doi: 10.1186/s12985-023-02034-x.

34. Zaki BM, Fahmy NA, Aziz RK, Samir R, El-Shibiny A. Characterization and comprehensive genome analysis of novel bacteriophage, vB_Kpn_ZCKp20p, with lytic and anti-biofilm potential against clinical multidrug-resistant *Klebsiella pneumoniae*. *Front Cell Infect Microbiol*. 2023 Jan 23;13. doi: 10.3389/fcimb.2023.1077995.
35. Baqer AA, Fang K, Mohd-Assaad N, Adnan SNA, Nor NS. In Vitro Activity, Stability and Molecular Characterization of Eight Potent Bacteriophages Infecting Carbapenem-Resistant *Klebsiella pneumoniae*. *Viruses*. 2022 Dec 30;15(1):117. doi: 10.3390/v15010117.
36. Laforet F, Antoine C, Reuter BB, Detilleux J, Pirnay JP, Brisse S, et al. In Vitro and In Vivo Assessments of Two Newly Isolated Bacteriophages against an ST13 Urinary Tract Infection *Klebsiella pneumoniae*. *Viruses*. 2022 May 1;14(5). doi: 10.3390/v14051079.
37. Feng J, Li F, Sun L, Dong L, Gao L, Wang H, et al. Characterization and genome analysis of phage vB_KpnS_SXFY507 against *Klebsiella pneumoniae* and efficacy assessment in *Galleria mellonella* larvae. *Front Microbiol*. 2023 Jan 30;14:1081715. doi: 10.3389/fmicb.2023.1081715.
38. Junqueira JC, Mylonakis E, Borghi E. *Galleria mellonella* experimental model: Advances and future directions. *Pathog Dis*. 2021 Apr 22;79(5):ftab021. doi: 10.1093/femspd/ftab021.
39. Mikulak E, Gliniewicz A, Przygodzka M, Solecka J. *Galleria mellonella* L. as model organism used in biomedical and other studies. *Przegl Epidemiol*. 2018;72(1):57-73. PMID: 29667381.
40. Nazir A, Qi C, Shi N, Gao X, Feng Q, Qing H, et al. Characterization and Genomic Analysis of a Novel Drexlervirial Bacteriophage IME268 with Lytic Activity Against *Klebsiella pneumoniae*. *Infect Drug Resist*. 2022;15:1533–46. doi: 10.2147/IDR.S347110.
41. Gorodnichev RB, Kornienko MA, Malakhova MV, Bespiatykh DA, Manuvera VA, Selezneva OV, et al. Isolation and Characterization of the First Zobellviridae Family Bacteriophage Infecting *Klebsiella pneumoniae*. *Int J Mol Sci*. 2023 Feb 1;24(4). doi: 10.3390/ijms24044038.
42. Liang B, Zhao W, Han B, Barkema HW, Niu YD, Liu Y, et al. Biological and genomic characteristics of two bacteriophages isolated from sewage, using one multidrug-resistant and one non-multidrug-resistant strain of *Klebsiella pneumoniae*. *Front Microbiol*. 2022 Oct 13;13:943279. doi: 10.3389/fmicb.2022.943279.
43. Hao G, Shu R, Ding L, Chen X, Miao Y, Wu J, et al. Bacteriophage SRD2021 recognizing capsular polysaccharide shows therapeutic potential in serotype K47 *Klebsiella pneumoniae* infections. *Antibiotics (Basel)*. 2021 Jul 22;10(8):894. doi: 10.3390/antibiotics10080894.

Figure 1. Morphology of phage GA23 on LB agar. A translucent halo surrounds each lytic plaque after overnight culture. Scale bar = 1 cm.

Figure 2. (A) pH stability of phage GA23 after incubation at different pH for 1 hour. (B) Thermal stability of phage GA23. Lytic activity was lost when GA23 was incubated at 50°C for 1 hour.

Figure 3. One-step growth curve of phage GA23. Exponential phase lasted 30 minutes, with a burst size of 161 viral particles/cell.

Figure 4. Percentage of survival of *G. mellonella* larvae infected with ESBL-producing *K. pneumoniae*. Each group comprised 15 larvae. Notably, groups inoculated with phage, SM-buffer, or both maintained 100% survival. Only 26.7% of non-treated larvae survived after 96 hours.

Strain	Spot Test
ESBL	Susceptible
VIM-1	Non-susceptible
IMP-8	Non-susceptible
NDM-1	Non-susceptible
KPC-2	Non-susceptible
OXA-48-like	Non-susceptible
OXA-181	Non-susceptible

Table 1. Spot testing of multidrug-resistant *K. pneumoniae* strains with phage GA23. Abbreviation: ESBL (Extended Spectrum Beta Lactamase)

Address correspondence to:

Gustavo Alonso Quispe Villegas, MD

Universidad Peruana Cayetano Heredia

Av Honorio Delgado 430, SMP

Lima, Peru

Email: gustavo.quispe.v@upch.pe

Dear Dr. Christina Cuomo,
Editor-in-Chief
Microbiology Spectrum – ASM

November 29, 2024

I hope this message finds you well.

Regarding our manuscript (code: Spectrum 01145-24), I am pleased to inform you of the submission of the revised version of our manuscript, titled: “In vivo evaluation of phage therapy against *Klebsiella pneumoniae* using the *Galleria mellonella* model and molecular characterization of a novel *Drulisvirus* phage species”.

In response to the reviewers comments, this version addresses all issues assessed point-by-point:

I would suggest authors revise the manuscript and improve their writing.

- The manuscript writing and grammar of the manuscript have been improved.

Results and discussion, which are the essence of the paper, are poorly written, and no proper analysis was done.

- The results and discussion section has been rewritten, including the WGS analysis.

The abstract needs to be rewritten, especially the conclusion.

- The abstract has been rewritten to align with the modified manuscript version (including the conclusion).

In the plaque isolation heading, why was plaque picked up in BHI medium but not in phage buffer?

- We rechecked our protocols; we finally used LB medium.

What is the reason behind the 1-hour incubation of phage plaque in BHI? Doing so, what author wanted to check?

- We incubated our phage plaque for 1 hour to allow the phage particles to diffuse out of the agar, according to Jamalludeen, et al. (2007)

What was the control for pH and thermal stability? Statistics need to be done in the pH and thermal stability experiment by selecting any control temperature and pH value. Additionally, statistical tests can be applied to determine significant changes in titer with respect to control.

- We did not use controls; however, we performed a Kruskal-Wallis test with a post-hoc Dunn test to determine significant changes in titers in both experiments. We concluded phage titers are not statistically different between pH values.

Phage pH stability could be evaluated at a higher alkaline pH (9-14) as well.

- Our pH stability testing was conducted to assess the suitability of the phage in physiological conditions. For this reason, we did not evaluate our phage at higher alkaline pH levels.

Thermal stability was measured at a 1-degree change. It is obvious that a change in titer cannot be observed; the temperature range can be increased by at least 5 degrees.

- For the reasons mentioned above, our assessment evaluated our phage under physiological conditions, including temperature. We also included temperatures for refrigeration and under febrile scenarios.

Eclipse (if chloroform was added to the aliquot at each interval) and latent period can be found by a one-step growth curve, which is not calculated. Kindly add a latent period for phage.

- Our protocol did not include chloroform. Moreover, we estimated our latent period to be 5 minutes, since the latent period for our phage was very short, as reported for other Drulisvirus according to Hao, et al. (2021) and Laforet et al. (2022).

For phage therapy, phenotypic and genotypic characterization is necessary, which determines the phage family and safety on a genotypic level. Why were these experiments not performed?

- We performed WGS and determined genome content, gene annotation, and phage taxonomy.

In Fig. 1, the morphology of phage GA23 seems to be not a clonal plaque. There are different plaques with variable sizes. A smaller halo plaque is different from other plaques. Kindly update the manuscript with a plaque plate that shows uniform morphology.

- The figure 1 was updated with another plaque plate. Clonality was demonstrated with WGS, which resulted in only one phage.

Our revised manuscript also incorporates Reviewer 2 observations.

We believe that the information presented could be of great interest to SPECTRUM. Additionally, I would like to reiterate that this article has not been submitted to any other journal.

We thank the reviewers for their constructive comments and valuable suggestions, which have greatly contributed to improving the quality of our manuscript.

Sincerely,

Jesús Tamariz

Universidad Peruana Cayetano Heredia

Re: Spectrum01145-24R1 (*In vivo* evaluation of phage therapy against *Klebsiella pneumoniae* using the *Galleria mellonella* model and molecular characterization of a novel *Drulisvirus* phage species)

Dear Dr. Jesus Humberto Tamariz:

Thank you for the privilege of reviewing your work. Below you will find my comments, instructions from the Spectrum editorial office, and the reviewer comments.

Revision Guidelines

Sincerely,
Bobby Warren
Editor
Microbiology Spectrum

Reviewer #1 (Comments for the Author):

The author addresses the comments well but still few points still need to be clarified before the final decision:

1. What will be its potential against other obtained or isolated carbapenemase-producing strains of *Klebsiella* or its activity is a too-narrow spectrum and related to one strain only?
2. The inactivation of the phage was observed at 50 degrees centigrade which makes it a weak candidate as compared to other

isolated phages with a higher temperature range.

3. Highlight bacterial resistance to phages in terms of your analysis.

4. Do cocktail studies synergize their activity against broad host ranges?

Dear Dr. Christina Cuomo,

Editor-in-Chief

Microbiology Spectrum – ASM

February 24, 2025

I hope this message finds you well.

Regarding our manuscript (code: Spectrum 01145-24R1), I am pleased to inform you of the submission of the revised version of our manuscript, including the clarifications requested by Reviewer 1.

In response to the reviewer comments:

1. What will be its potential against other obtained or isolated carbapenemase-producing strains of *Klebsiella* or its activity is a too-narrow spectrum and related to one strain only?
 - The isolated phage is specific to a single bacterial strain. As mentioned in the manuscript, our host range analysis against six carbapenemase-producing *K. pneumoniae*. However, this finding does not diminish its therapeutic potential, as ESBL-producing *Enterobacteriaceae* continue to pose a significant public health concern, contributing to increased rates of treatment failure and healthcare costs, according to the WHO. The manuscript discussion has been modified to incorporate this statement.
Reference: World Health Organization. WHO bacterial priority pathogens list, 2024: bacterial pathogens of public health importance to guide research, development and strategies to prevent and control antimicrobial resistance [Internet]. Geneva: World Health Organization; 2024 [cited 2025 Feb 21]. Available from: <https://iris.who.int/bitstream/handle/10665/376776/9789240093461-eng.pdf>
2. The inactivation of the phage was observed at 50 degrees centigrade which makes it a weak candidate as compared to other isolated phages with a higher temperature range.
 - Inactivation at 50°C does not diminish its lytic potential, as such temperatures are not encountered during experimental procedures nor in potential therapeutic applications of the phage. Additionally, resistance to temperatures of 60–70°C would provide no therapeutic benefit, as these conditions are incompatible with human life.
3. Highlight bacterial resistance to phages in terms of your analysis.

- Evaluating bacterial resistance to phages was not the objective of this study. However, it is well established that bacteria naturally develop resistance to bacteriophages. In the *Klebsiella* genus, preventing phage adsorption is the primary resistance mechanism, attributed to the diversity of capsular antigens and their steric role in masking bacterial surface receptors. Additionally, intracellular antiviral strategies have been described, including the restriction-modification (R-M) system, the toxin-antitoxin (TA) system, and the CRISPR-Cas system. Characterization of these phage resistance mechanisms are beyond the scope of our study.
- References:
 - Dunstan RA, Bamert RS, Tan KS, Imbulgoda U, Barlow CK, Tairaoa G, et al. Epitopes in the capsular polysaccharide and the porin OmpK36 receptors are required for bacteriophage infection of *Klebsiella pneumoniae*. *Cell Rep.* 2023 Jun 27;42(6).
 - Labrie SJ, Samson JE, Moineau S. Bacteriophage resistance mechanisms. Vol. 8, *Nature Reviews Microbiology*. 2010. p. 317–27.
 - Cai R, Wang G, Le S, Wu M, Cheng M, Guo Z, et al. Three capsular polysaccharide synthesis-related glucosyltransferases, GT-1, GT-2 and WcaJ, are associated with virulence and phage sensitivity of *Klebsiella pneumoniae*. *Front Microbiol.* 2019;10(MAY).
 - Egado JE, Costa AR, Aparicio-Maldonado C, Haas PJ, Brouns SJJ. Mechanisms and clinical importance of bacteriophage resistance. Vol. 46, *FEMS Microbiology Reviews*. Oxford University Press; 2022.

4. Do cocktail studies synergize their activity against broad host ranges?

- The use of phage cocktails to enhance activity against a broader host range is well documented. However, this study examines the properties of an isolated phage. Rather than identifying additional phages, our objective was to characterize its microbiological and molecular properties and assess its therapeutic potential using *Galleria mellonella* as an *in vivo* model.
- References: Chen H, Liu H, Gong Y, Dunstan RA, Ma Z, Zhou C, et al. A *Klebsiella*-phage cocktail to broaden the host range and delay bacteriophage resistance both *in vitro* and *in vivo*. *NPJ Biofilms Microbiomes*. 2024 Nov 14;10(1):127. doi: 10.1038/s41522-024-00603-8.

We thank the reviewers for their constructive comments and valuable suggestions, which have greatly contributed to improving the quality of our manuscript.

Yours faithfully,

Jesús Tamariz

Universidad Peruana Cayetano Heredia

Re: Spectrum01145-24R2 (*In vivo* evaluation of phage therapy against *Klebsiella pneumoniae* using the *Galleria mellonella* model and molecular characterization of a novel *Drulisvirus* phage species)

Dear Dr. Jesus Humberto Tamariz:

Your manuscript has been accepted, and I am forwarding it to the ASM production staff for publication. Your paper will first be checked to make sure all elements meet the technical requirements. ASM staff will contact you if anything needs to be revised before copyediting and production can begin. Otherwise, you will be notified when your proofs are ready to be viewed.

Sincerely,
Bobby Warren
Editor
Microbiology Spectrum